# Oxidized Carbon-Based Spacers for Pressure-Resistant Graphene Oxide Membranes

**DOI:** 10.3390/membranes12100934

**Published:** 2022-09-26

**Authors:** Ekaterina A. Chernova, Konstantin E. Gurianov, Dmitrii I. Petukhov, Andrei P. Chumakov, Rishat G. Valeev, Victor A. Brotsman, Alexey V. Garshev, Andrei A. Eliseev

**Affiliations:** 1Department of Materials Science, Lomonosov Moscow State University, 1-73 Leninskiye Gory, Moscow 119991, Russia; 2Department of Chemistry, Lomonosov Moscow State University, 1-3 Leninskiye Gory, Moscow 119991, Russia; 3ESRF—The European Synchrotron Radiation Facility, 71, Avenue des Martyrs, 38043 Grenoble, France; 4Udmurt Federal Research Center of the Ural Brunch of Russian Academy of Sciences (UdmFRC of UB RAS), St. Them. Tatiana Baramzina 34, Izhevsk 426067, Russia

**Keywords:** graphene oxide, fullerenols, graphene oxide nanoribbons, pressure stability, dehumidification, water vapor transport

## Abstract

In this study, we report the influence of carbon-based spacer-oxidized derivatives of fullerenes (fullerenols) C_60_(OH)_26–32_ and graphene oxide nanoribbons on the performance and pressure stability of graphene-oxide-based composite membranes. The impact of the intercalant shape and composition on the permeance of the selective layers for water vapors has been studied under pressure gradients. It is shown that the insertion of ball-shaped fullerenols between graphene oxide nanoflakes allows a suppression in irreversible permeance loss to 2–4.5% and reversible permeance loss to <25% (at 0.1 MPa), while retaining large H_2_O/N_2_ selectivities of up to ~30,000. The demonstrated approach opens avenues for the highly effective stabilization of GO membranes at elevated pressures for industrial-scale dehumidification.

## 1. Introduction

Graphene oxide (GO) has gained attention as a promising membrane material due to its cost-effective synthesis, ease of membrane assembly on various supports, diverse chemical nature of functional groups, and reasonable chemical stability [1,2]. The hydrophilic nature of GO gives rise to the high performance of the membranes in water-penetration processes [3,4]. The outstanding performance of GO membranes has been demonstrated in pervaporation [5], reverse osmosis [6], and air dehumidification processes [7]. However, two divergent processes caused by weak bonding between GO nanosheets and their flexible nature–swelling in water solutions and compaction under external pressure prevent their industrial application. Swelling in water or polar organic solvents results in an increase in interlayer distances, leading to a decrease in rejection coefficients and even results in membrane disintegration [8,9]. The problem of swelling has been successfully solved by the chemical cross-linking of nanoflakes or physical intercalation of stabilizer agents with no chemical bonding to GO nanosheets or a combination of both approaches [10]. As a result, the enhancement of GO membrane long-term stability up to 120 h in pervaporation process can be achieved [5].

Membrane compaction is much rarely discussed than swelling. The problem is associated with a decrease in interlayer spacing in pressure-driven processes, resulting in a flux drop of over 10 times during water filtration [11,12] and significant reductions in water-vapor permeance in air dehumidification [13]. While GO membranes operating without transmembrane pressure demonstrate ultimate water vapor permeances of up to 100 m^3^(STP)∙m^−2^∙bar^−1^∙h^−1^ with a H_2_O/N_2_ selectivity of ~10,000 [7], the membranes operating at a transmembrane pressure of ~1 bar exhibit a water-vapor permeance in the range of 7–9 m^3^(STP)∙m^−2^∙bar^−1^∙h^−1^ with a H_2_O/N_2_ selectivity of ~1000 only [14]. An increase in transmembrane pressure of up to 4 bar leads to a further suppression of water-vapor permeance by 2–3 times [14]. Strong deviations from the Darcy law is also characteristic for pressure-driven liquid-water filtration [15]. 

From a practical point of view, the operation under pre-defined elevated feed stream pressure during the dehumidification process is required due to the possibility of using a part of a dehumidified retentate stream after decompression as a self-sweep flow. Otherwise, in the absence of a pressure difference between the feed and permeate sides in a membrane dehumidifier, an external sweep flow with low humidity is required [16].

To prevent graphene oxide compaction under external pressure, approaches associated with the creation of elastic incompressible channels between GO nanoflakes are considered. For this purpose, various interflake spacers have been used, including fullerenes [15], single-walled carbon nanotubes [17], Cu(OH)_2_ nanostrands [18], zeolitic imidazolate framework particles [19,20], and graphitic carbon nitride layers (g-C_3_N_4_) [21]. Those allowed the stabilization of GO interlayer spacing and retaining water permeances of ~60 m^3^∙ m^−2^ ∙h^−1^ ∙bar^−1^ under the transmembrane pressure of 1 bar [20]. Among a variety of spacers, carbon-based nanomaterials and their derivatives seem favorable for intercalations between GO layers due to a strong affinity towards graphene oxide. The functionalization of these spacers with polar groups enables further binding to GO. Recently, ethylene-diamine-functionalized (C_60_(EDA)_10_) [22] and pyrrolidinium-functionalized (C_60_(Py)^n+^) C_60_ fullerenes [23] have been introduced into the interlayer space of graphene oxide to create nanogalleries with a height of 15.6 Å for improved water transport. This technique allowed an increase in water permeance by 5–7 times compared with pristine graphene oxide. However, being focused on swelling suppression, the authors omitted the evaluation of compaction effects under elevated transmembrane pressure. 

Despite the concept of stabilization of GO structure by encapsulation of spacers that is considered as generally effective, an understanding of the role of spacers geometry is still lacking due to the absence of a direct comparison of the resulting structures and the membrane’s performance. Thus, here, we focused on the preparation and transport characteristics of composite membranes based on the GO nanoflakes encapsulated ball-shaped fullerenols and belt-shaped GO nanoribbons. A direct deposition from nanoflakes/encapsulant colloidal mixtures was employed in both cases. The influence of the morphology and composition of the spacers on the microstructure, transport characteristics, and pressure stability of water-vapor permeance of the GO composite membranes is considered.

## 2. Experimental Part

### 2.1. Preparation of Oxidized Graphitic Carbon Precursors 

Suspensions of GO nanoflakes were obtained by an improved Hummers’ method according to the procedure described in [24] using a graphite:KMnO_4_ ratio of 1:6 [25]. The purification of the suspensions from residual inorganic ions was performed by dialysis for 30 days. The extent of sulfate-ions removal from the GO suspensions was controlled by XPS. The resulting graphene oxide sample was denoted as MFGO (medium-flake GO). As shown in detail in our previous paper, MFGO nanosheets have rectangular geometries with an average lateral size of 730–750 nm. [7]. Graphene oxide nanoribbons were obtained by the oxidation of pure single-walled carbon nanotubes (CNTs) with a similar method [26], resulting in nanoribbons with belt-like geometry, an average length of 140 nm, and a width of 10 nm [7]. The sample was denoted as CNTGO (carbon nanotube-based graphene oxide).

The synthesis of C_60_(OH)_n_·H_2_O_m_ fullerenol was performed according to the modified procedure reported in [27,28] using fullerene oxidation with 30% H_2_O_2_ in an alkali medium. Fullerene C_60_, purchased from Fullerene center (99.98 %) was used as a precursor. A solution of fullerene (200 mg, 0.28 mmol) in 200 mL of toluene was contacted with KOH solution (940 mg, 16.8 mmol, and 60-fold excess) in 100 mL of 30% H_2_O_2_ with an addition of 1 mL 40 wt.% tetrabutylammonium hydroxide (TBAH) as a phase-transfer catalyst. The mixture was vigorously stirred at 60 °C for 24 h. The color of toluene phase changed from deep purple to colorless, while the water phase turned light brown. Toluene phase was separated and compound was precipitated out as a light brown solid, washed with methanol solution to pH = 7, dried in vacuum, and dissolved in deionized water for further use. The product yield exceeded 90%. Fullerenols were characterized using FT-IR and UV-vis spectroscopy and TGA-MS/DSC and DLS methods (see Appendix A for details). According to the analyses, the fullerenol phase, C_60_(OH)_n_·H_2_O_m_, contains 26–32 -OH groups on the carbon cage (n = 26–32) and 1–3 water molecules per cage (m = 1–3). Furthermore, throughout the paper, the formula of fullerenol is expressed as C_60_(OH)_26–32_, omitting water content as a matter of convenience. According to the dynamic light scattering (DLS), the size of fullerenols is 1.7 ± 0.1 nm (as is presented in Appendix A). Note that this value reflects the maximum size of fullerenols as provided by the slipping plane in the solution.

### 2.2. Preparation of Composite Membranes

Porous anodic aluminum oxide (AAO) membranes with a pore diameter of ~80 nm and a thickness of 100 µm were used as supports for the deposition of thin selective layers. Porous supports were prepared by anodic oxidations of high-purity aluminium in 0.3 M H_2_C_2_O_4_ at 120 V. To minimize bending loads and the deformation of 2D selective layers with transmembrane pressure and to distribute thin-layer bearing areas, a supporting AAO layer with nanochannels branched to 40 nm was formed at the top of AAO membranes by variations in the anodic oxidation’s voltage [29]. A detailed description of the AAO preparation and characterization of their gas permeance is provided in [30,31]. 

To prepare composite membranes, the initial suspensions of MFGO, CNTGO, and fullerenols were diluted with deionized water and methanol (99% purity) to obtain stock suspensions with a concentration of 1.0 mg/mL and a H_2_O:CH_3_OH ratio of 1:1. Afterwards, GO suspensions were mixed with CNTGO or fullerenols to obtain mixtures for deposition on AAO substrates. To verify the influence of the amount of fullerenols on MFGO membrane permeance, mixtures containing 20 wt.% and 33 wt.% of fullerenols were deposited. The suspensions were first dropped (140 μL) and then spin-coated onto porous AAO supports under slight vacuum suction (~30 kPa) at 1500 rpm.

### 2.3. Characterization of Composite Membranes

Overview and region C1*s* and O1*s* X-ray photoelectron (XPS) spectra of the samples were collected on SPECS instrument (Specs GmbH, Berlin, Germany) using MgK-α excitation (E_ex_ = 1254 eV). The recorded spectra were calibrated to pure graphite C1*s* energy (284.6 eV). XPS data were treated with CasaXPS software package (version 2.3.24, Casa Software Ltd., Teignmouth, United Kingdom). Shirley type background and mixed Gauss (70%)-Lorentz (30%) functions were used for the deconvolution of spectra under fixed FWHM for all spectral components.

Scanning electron microscopy (SEM) observations were carried out using an Carl Zeiss Nvision 40 (Carl Zeiss Microscopy GmbH, Jena, Germany) instrument. The detailed analysis of selective layer cross-section was performed by transmission electron microscopy (TEM) using Libra 200 (Carl Zeiss, Microscopy GmbH, Jena, Germany) chromatic aberration-corrected electron microscope equipped by an Ω-filter and EELS and EDX X-Max 80T detector. A common procedure for FIB sample preparation using FEI Quanta 200 3D dual beam instrument (FEI (Thermo Fisher Scientific, Hilsboro, OR, USA)was used for cutting out thin (10–20 nm) cross-sectional specimens of composite membranes. 

Raman spectroscopy studies were performed on Renishaw InVia spectrometer equipped with Leica DMLM optics (50× objective) using an excitation wavelength of 633 nm (20 mW, He-Ne laser). Raman maps were acquired in the Streamline accumulation mode with a line-focused laser beam (~0.5 × 50 μm), providing a point-to-point resolution of 1.2 μm. The spectra were registered using 1200 L/mm gratings and Peltier-cooled 1024 × 568 detector, resulting in a spectral resolution of ~1 cm^−1^. The point exposure time equaled 200 s at maximum power input of <0.1 mW/μm^2^. Experimental data was processed using Wire 3.4 Renishaw software. (the Raman spectroscopy equipment and software was produced by Renishaw plc, Wotton-under-Edge, Gloucestershire, United Kingdom) The spectra were fitted by Pseudo-Voigt functions positioned at 1340 ± 20 and ~1590 ± 10 cm^−1^ for the D- and G-bands, respectively. The fits were performed in the range of 1250–1750 cm^−1^ using 2-rd order baseline subtractions and a final tolerance factor for each spectrum of 0.01.

The interlayer distance of composite membranes under the controlled humidity was measured with an ID10 beamline at the European Synchrotron Radiation Facility (ESRF, Grenoble, France) with respect to grazing incidence geometry (several beamtimes over July 2018–February 2022). An incident photon beam with an energy of 22 keV (λ = 0.564 Å, Δλ/λ ~ 1·10^−4^) was used. The beam was focused at the sample’s position by beryllium compound refractive lenses at a size of 250 × 54 μm^2^ (horizontal × vertical). A grazing incidence angle of ~0.10° was selected to maximize scattering intensity from the samples. Scattered X-rays were captured in GIWAXS geometry using a Pilatus 300 K 2D detector (pixel size 172 × 172 μm^2^) (**DECTRIS AG, Baden-Daettwil, Switzerland**). The sample-to-detector distance was set to 350 mm. A specially designed cell with Kapton windows was used to control humidity levels during the experiments [13]. A total flow of 1000 mL/min of mixed dry and wet (close to 100% humidity obtained by bottle-type humidifier) nitrogen was introduced to the cell with SLA5850 (Brooks Instrument, Hatfield, PA, USA) flow controllers. The humidity level was monitored with HIH-4000 (Honeywell) (Charlotte, NC, USA) sensors before and next to the experimental cell. Additionally, the spectra of wetted samples were acquired by placing a water drop onto the sample’s surface under >98% humidity. GIWAXS data were analyzed using the DPDAK software package (DESY (Hamburg) and MPIKG (Potsdam-Golm), Germany) [32].

Water sorption capacities were measured using a gravimetric technique that is described in detail in our previous paper [25]. 

Due to the barrier properties of graphene oxide layers towards permanent gases, the integral scheme of gas permeance measurements was employed for the characterization of GO permeability towards H_2_, He, CH_4_, N_2_, O_2_, CO_2_, C_4_H_10_, and SF_6_ [33]. A measurement cell consisting of a feed chamber and calibrated permeate chamber was used. Both chambers were evacuated down to a residual pressure lower than 0.1 mbar and then were disconnected by ball valves from the pump. Then, the gas stream was introduced into the feed chamber of the cell and, as the gas flux was permeating through the membrane, a gradual pressure increase with respect to the gas inside the permeate chamber was recorded in the form of a pressure-time dependence. The same scheme was used to study the transportation of water vapor in a dynamic mode. In this case, a wet nitrogen flow with nearly 100% humidity was supplied as a feed stream, and a pressure increase inside the permeate chamber was recorded until a pressure level exceeding saturated H_2_O vapor pressures (~5 kPa) was achieved (see Appendix A). The water vapor flux through the membrane was calculated using a tangent of the linear section slope of the pressure–time curve (curve section b, marked with red in Appendix A)

Measurements of water vapor permeance under steady state conditions were also performed according to the protocol described earlier in [7]. The feed side of the membrane was blown by gas with a controlled humidity obtained by intermixing dry and humid gas streams. The permeate side of the membrane was swept by He fluxes. The temperature and humidity of both gas fluxes were controlled by HIH-4000 sensors (Honeywell, USA). The dew point temperature of the outlet gas flux was also determined by a dew-point hygrometer TOROS-3VY (Gas Institute of the Academy of Sciences of Ukraine, Kyiv, Ukraine). Feed and permeate pressures were equal to 1 atm. All experiments were carried out at temperatures in the range of 23–25 °C. 

The pressure stability of the membranes was studied under a stepwise increase in feed pressure (with the step of 0.2 bar) followed by prolonged exposure (typically >2 h) to a pressure gradient until a constant equilibrium in water-vapor flux was achieved. Finally, the pressure gradient was released with the same step of 0.2 bar. During the experiment, the partial pressure of water vapors in the feed stream was kept constant (3158 Pa at 25 °C). 

## 3. Results and Discussion

### 3.1. Microstructure of the Membranes

The model’s ideal microstructures of the composite membranes are schematically represented in Figure 1a,b: Graphene oxide nanoribbons and fullerenols are expected to be distributed between MFGO nanoflakes to form flexible pressure-stable nanogalleries. According to SEM results, the insertion of both fullerenols and graphene oxide nanoribbons between MFGO nanoflakes does not induce any significant changes in the texture and microstructure of the composite membranes. The selective layers with inherent wrinkles are spreading uniformly on the porous AAO substrates (Figure 1c,d). The HRTEM of membrane cross-sections reveals disordered lamellar structures of the selective layers (Figure 1e,f) with a thickness uniformity proved by C-K maps (Figure 1g,h). The encapsulation of fullerenols between MFGO nanosheets is manifested by bright point-like inhomogeneities that are visible in energy filtered carbon edge maps (shown with red arrows in the Figure 1h). Indeed, according to the semi-empirical models of GO [25] and fullerenols, the latter express much higher local atomic densities of carbon atoms (>80 atoms/nm^3^) compared to that of dry GO (~50 atoms/nm^3^). Increased local atomic densities will result in excessive electron absorption/scattering by carbon atoms, which can be detected with energy-filtered carbon K edge TEM. Unfortunately, the insertion of nanoribbons cannot be distinguished unambiguously in TEM/EFTEM images due to their flat shape and nearly similar chemical compositions compared with MFGO flakes (see XPS results). 

According to SEM micrographs, the thickness of the selective layers is in the range of 30–45 nm (Table 1, Figure 1c,d). This thickness is consistent with the typical values for GO films prepared by spin-coating from water–methanol suspensions [7]. The difference between the observed thickness in SEM cross sections and TEM cuts corresponds, probably, to the fluffing of GO selective layers at the edges when the cross-sections of the samples were prepared. As a result, the observed layer thickness in SEM increases effectively. 

The uniformity of the selective layers was also confirmed by Raman scattering maps (Appendix A and Table 1). The intensity distribution along the membrane area stays within 25%, suggesting rather a homogeneous coating. In addition, the averaged Raman intensity allowed us to estimate the porosity of the selective layers according to a protocol reported earlier in [7]. Indeed, about a twice lower Raman intensity was detected for CNTGO@MFGO membranes (at a similar thickness of the selective layers), suggesting a strong density change in the coating (Table 1). In contrast, the estimate for C_60_(OH)_26–32_@MFGO membranes indicated no additional pores within the selective layers. As the result seems rather surprising, we can also attribute the effect to the difference in the contribution of D- and G-modes to Raman scattering intensities from ball-shaped fullerenols (Appendix A).

The chemical composition of the selective layers was analyzed by the XPS method with standard deconvolution of the peaks in C1s and O1s spectral regions (Figure 2 and Appendix A). According to the XPS data, the reference MFGO membrane exhibits an oxidation degree specific for graphene oxide obtained using the graphite:KMnO4 ratio of 1:6 in the improved Hummers’ method [7]. Both pure fullerenol and CNTGO phases reveal the prevalence of C-C bonds, covering 50% of total carbon intensity. In the case of fullerenols, it suggests that ~30 carbon atoms of C_60_ cage are involved in binding to oxygen-containing functional groups. Notably, the presence of a rather large amount of carbonyl groups could be considered as a sign for overoxidation accompanied by some destruction of C_60_ cages [34]. Graphene oxide nanoribbons also reveal lower oxidation degrees compared to MFGO. This can be ascribed to easier OH-group migrations to the edges of graphene flakes, oxygen association, and GO reduction in vacuum conditions [35]. GO nanoribbons also contain a slightly higher portion of C=O groups in comparison to MFGO, manifesting larger contributions from GO edges. Slight shifts in the resolved band positions of <0.15 eV appearing upon the inclusion of nanoribbons and fullerenols into GO lay below the energy resolution of the XPS spectra (~0.2 eV) and cannot be reliably ascribed to any chemical changes in the structure.

### 3.2. Nanoflakes Arrangement and Mass-Transport Properties of the Membranes

The permeance of composite membranes strongly relates to the size of interlayer galleries; thus, a detailed examination of layer arrangement and d-spacing dynamics was performed in the entire humidity range using the GIWAXS method (Figure 3a, Appendix A). The average size of interlayer galleries in the reference MFGO membrane expectedly changes from 0.7 nm for nearly dry sample to 1.2 nm for the sample saturated at 100% RH, as it was previously reported in [25,36]. The insertion of carbon spacers between MFGO nanoflakes leads to a slight increase in d-spacing in the entire range of RH. The effect was much better pronounced for fullerenol spacers, where d-spacing attained 1.4 nm at RH~100%. The observed increase in d-spacing after fullerenols intercalation between graphene oxide nanosheets follows the data reported earlier in [22], in which the authors observed that introducing functionalized fullerene C_60_(EDA)_10_ between graphene oxide nanosheets resulted in a d-spacing increase from 0.9 nm to 1.12 nm. Besides the larger interlayer spacing, the insertion of species also leads to an increased FWHM of the diffraction maxima, which is indicative for enhanced corrugations with respect to the layers (Figure 3d). A significant increase in FWHM is observed with the insertion of carbon spacers, manifesting the disordering of GO layers. For instance, a 3.5-fold and 10-fold increase in FWHM was registered with the insertion of fullerenols between GO nanoflakes in dry and wet states, respectively (Figure 3d, Appendix A). 

Structural disorders and increased void volumes between the layers should obviously affect the sorption characteristics of composite membranes. Water sorption isotherms (Figure 3b) even reveal the lower capacity of composite membranes compared to pure MFGO at low humidity (<40%). Accounting for rather small d-spacings in these conditions, the observed result can be attributed to a volumetric contribution of the spacers and hindered transport inside GO nanogalleries. In contrast, at high humidity, the sorption capacity of pure GO is strongly exceeded by the sorption capacity of nanoribbons- and fullerenol-encapsulated samples. We suspect that the spreading effect from the spacers results in increased spacings between nanoflakes, with a corresponding elevation in water-vapor content in the structure.

To understand the influence of the observed structural and sorption changes caused by spacers insertion on the transport properties of the selective layers, we proceed with an evaluation of membranes’ permeability towards permanent gases and water vapors depending on relative humidity and pressure conditions (Figure 3c,e,f). It was revealed that the permeance of composite membranes is greatly affected by the nature of the inserted species. Both nanoribbons and fullerenols increase the permeance of the MFGO barrier towards permanent gases. However, while the parasitic permeance for fullerenol-encapsulated MFGO increased by ~30% compared to the dense MFGO layer, encapsulation with nanoribbons leads to a sudden increase in permeance for permanent gases at ~5 times. This is attributed to the much higher porosity and layer disorder appearing due to an induction of extended and elongated nanoribbons. As a result, transport channels along nanoribbon edges accessible by permanent gases are created. Nevertheless, the permeance towards permanent gases for all composite membranes can be considered as barrier enough, and it is appropriate for membrane applications.

In contrast to permanent gases, the permeation of water vapors through composite membranes is several orders of magnitude faster. It is described with the capillary condensation of vapors and continuous medium flow within the galleries [7]. Despite the larger d-spacing and higher water sorption capacity of composite membranes, they show no enhancements in the transport of water vapors (Figure 3f, Table 2). Contrary, a ~20–30% decrease in absolute water-vapor permeance was observed. A similar decrease in liquid water flux was observed in [22] for GO selective layers intercalated with EDA-fullerenols, which is attributed to the jamming of excessive GO diffusion paths for water molecules. Indeed, water molecules travelling along the galleries can be effectively scattered on the encapsulated species, which leads to an elongation of the diffusion pathways and, consequently, reduces the diffusion coefficients. Thus, we consider that permeability losses are attributed to an increase in the length of diffusion pathways for water molecules (Figure 3d–f).

In order to elucidate the changes in travelling distance, diffusion coefficients were calculated using labyrinthine transport model [37]:(1)$$D_{labyrinth}=\frac{P}{h\cdot S}\frac{{L_{av}}^{2}}{4}\cdot \frac{l}{d_{int}}$$
(2)$$S=\frac{a\cdot \rho \left(GO\right)}{p}$$
where *S* denotes the sorption coefficient; *a* denotes the mass of water adsorbed by 1 g of a sample at a certain humidity; *ρ*(*GO*) denotes density of graphene oxide (1.8 g∙cm^−3^); *p* denotes partial water vapor pressure; *P* denotes water vapor permeance at a certain humidity; *l* denotes selective layer thickness; *d_int_* denotes interlayer thickness; *h* denotes slit thicknesses available for diffusion (0.35 nm); *L_av_* denotes the average lateral size of graphene oxide nanosheets (750 nm).

According to the calculations (Table 3), the insertion of carbon spacers inevitably decreases the effective diffusion coefficients for water molecules. More accurately, the change in diffusion pathways can be described by including the tortuosity of the channels. The latter increased at about twice the previous level with the encapsulation of fullerenols and by ~4 times with nanoribbons.

On the other hand, high absolute permeability values and RH-dependences of the permeance reveal that the capillary’s condensation remains the underlying mechanism for the transport of water vapors through the composite membranes. All membranes exhibit water-vapor permeances of 40–60 m^3^∙m^−2^∙bar^−1^∙h^−1^ at zero pressure drop and a relative humidity of the feed stream at ~90% with an ideal H_2_O/N_2_ selectivity in the order of 10^4^–10^5^ [7]. A large H_2_O/N_2_ selectivity of ~30,000 was retained with the encapsulation of fullerenols. The decrease in selectivity for the CNTGO@MFGO membrane obviously follows with increased parasitic N_2_ permeances introduced by nanoribbons.

While exhibiting lower water=vapor permeance at zero absolute pressure drop, the composite membranes illustrate much better stability towards membrane compaction under uniaxial loading. The permeance of the reference MFGO membrane decreases dramatically with the elevation of transmembrane pressure. The membrane loses over 60% of the initial permeance at 1 bar pressure drop (Figure 3f). Moreover, a loss of ~35% is irreversible, resulting in a water permeance of pressure-exposed membrane at ~40 m^3^∙m^−2^∙bar^−1^∙h^−1^. This loss is attributed to a compaction of the initially loosely packed GO nanoflakes of freshly prepared membrane into an ordered structure under a pressure gradient. The same effect was observed with 30-fold decrease in GO permeance for liquid water in the pressure-driven nanofiltration experiment, in which the authors attribute this compaction to the ordering of loosely packed GO nanoflakes [38]. In our opinion, such a strong irreversible compaction occurs due to chemical binding of GO flakes when no (or a single layer of) water is contained between the layers. The compaction can then expand in the lateral direction, further diminishing the permeance of GO membranes.

The insertion of carbon-based spacers predictably prevents the collapse of the GO selective layer, resulting in a much higher pressure stability and lower permeance losses. In the case of nanoribbons, a dynamic permeance drop decreases to ~50% with an irreversible loss of <10% (Figure 3f). Despite having similar absolute permeance compared to MFGO at elevated transmembrane pressures, this membrane has an advantage compared to pure GO membranes after the pressure release. It is obviously attributed to the formation of pressure-stable elastic channels between GO nanoflakes, enhancing the total membrane resistance towards pressure gradients. The pressure stability becomes even more pronounced with the insertion of fullerenols. The permeance loss at the transmembrane pressure of 1 bar in this case does not exceed 25%, while the irreversible permeance loss diminishes to ~5%. The performance of the fullerenol-containing membrane surpasses the reference membrane over the entire pressure range and its dominance grows with the transmembrane’s pressure. We consider that the corrugation of flakes appeared with the encapsulation of the fullerenol species, and this both prevented the crosslinking of GO layers and provided a necessary void spacing for the absorption of water. Compared to the extended nanoribbons, ball-shaped fullerenols enable a much lower tortuosity of the diffusion paths and provide lower parasitic porosities with resepct to the membranes, thus providing improved selectivities and performances (Figure 3g–i). Thus, the insertion of carbon spacers prevents the sticking of GO nanosheets, providing improved stabilities towards elevated pressure drops.

It should be noted that the amount of both spacers should definitely strongly influence the membrane’s structure and performance. The role of the amount of nanoribbons on the permeance of the membranes was reported earlier in Ref [39], revealing a strong enhancement in parasitic permanent gas flow when increasing the amount of nanoribbons. However, no permeance stabilizing effects were exposed in the work. The amount of fullerenols intercalated in GO can principally be varied from 0 wt.% (pristine graphene oxide membrane) to a certain amount providing the closest packing of fullerenols between GO nanoflakes. The maximum loading of fullerenols in GO, accounting for the fullerenol’s size of 1.2 nm, its molecular weight of 1230 g/mol, and graphene oxide density of 1.8 g/cm^3^, provides an estimate of ~50 wt.%. However, packing fullerenols too closely will obviously block the permeation channels for water molecules. The effect is well exposed when lowering water-vapor permeance and decreasing H_2_O/N_2_ selectivities while increasing the loading of spacers (Figure 3f, Table 2). On the other hand, increased loading enables the improved stabilization of GO structure with respect to both reversible and irreversible compaction. Optimal loading is considered to be dependent on the membrane’s application and the utilized pressure drops.

Thus, fullerenol spacers enable strong enhancements in the pressure resistance of GO-based membranes and perform as effective anti-compaction agents, allowing the reversible “breathing” of GO’s lamellar structure under non-zero pressure drops. Accounting for the ultra-low thickness of composite membranes and the decreasing market price of fullerenes and their derivatives, the proposed solution can be considered as commercially effective for implementations in the large-scale production of dehumidified membranes.

## 4. Conclusions

To summarize, the influence of morphology and composition of the carbon-based spacers on the microstructure, gas-transport characteristics, and stability of water vapors transport at elevated transmembrane pressure is explored. The ball-shaped fullerenols act as stable spacers for graphene oxide nanogalleries, providing both excellent resistance towards pressure gradients and lower parasitic gas permeance compared to belt-shaped nanoribbons. Medium-flake graphene oxide membranes encapsulated with 20 wt.% of C_60_(OH)_26–32_ fullerenols exhibit a permeance loss of <25% at the transmembrane pressure of 0.1 MPa, with a <5% irreversible losses compared to the ~3 fold decrease and over 35% irreversible losses in pure GO membranes. We believe that the proposed approach can be extended to enhance the pressure stability of 2D materials, providing reliable mechanical and long-term stability with respect to the membranes.

## Figures and Tables

**Figure 1 membranes-12-00934-f001:**
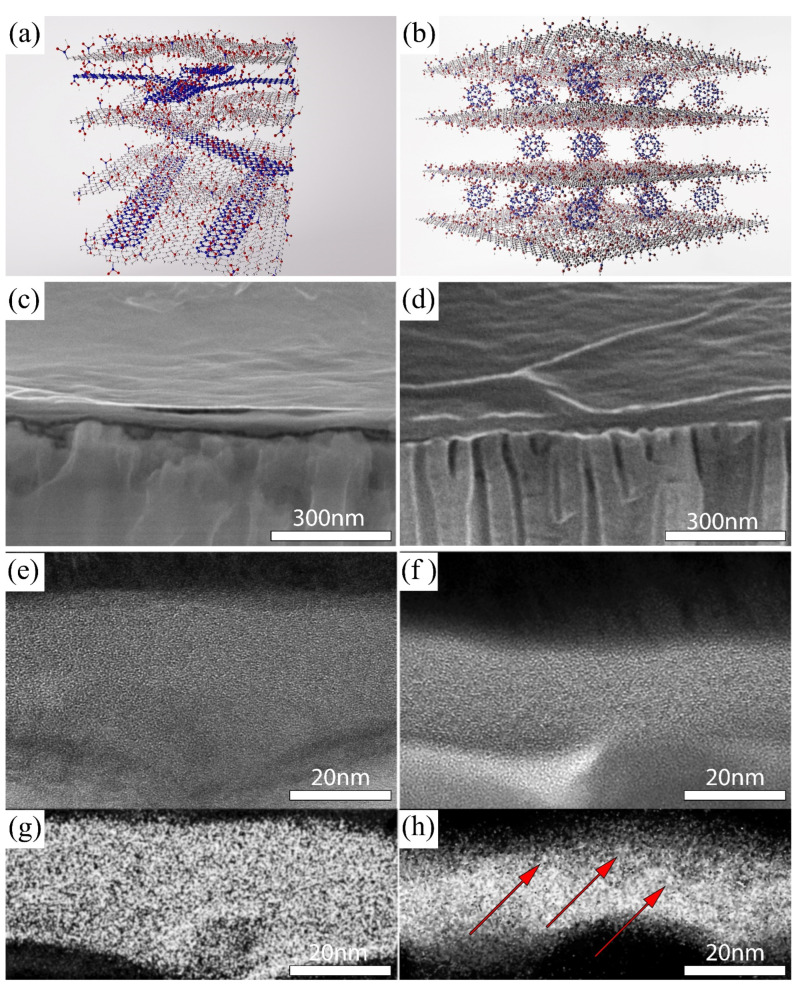
Schematic representation of microstructures of composite membranes: (**a**) CNTGO@MFGO and (**b**) C_60_(OH)_26–32_@MFGO; SEM and TEM micrographs of the cross-sections of the composite membranes: (**c**,**e**) CNTGO@MFGO and (**d**,**f**) C_60_(OH)_26–32_@MFGO; energy-filtered carbon K edge images of (**g**) CNTGO@MFGO and (**h**) C_60_(OH)_26–32_@MFGO samples. Due to an increased local atomic density of carbon atoms in fullerenols (>80 atoms/nm^3^) compared to GO (~50 atoms/nm^3^), they are seen on C1s EFTEM as brighter spots and inhomogeneities. Red arrows in (**h**) indicate the possible location of fullerenol inclusions.

**Figure 2 membranes-12-00934-f002:**
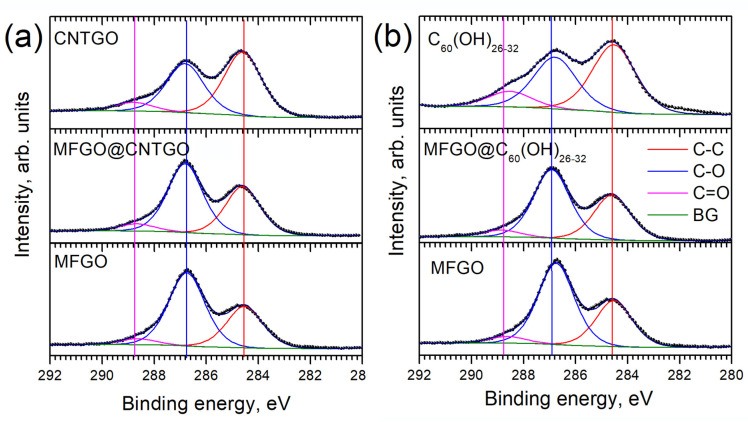
XPS spectra of composite membranes: (**a**,**b**)—C1s-spectra for GO nanoribbons- and fullrenol-intercalated samples, respectively.

**Figure 3 membranes-12-00934-f003:**
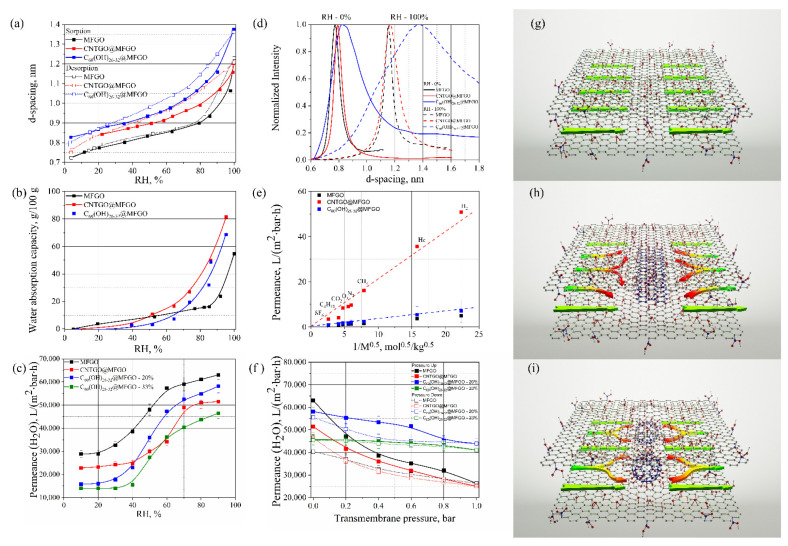
The relative humidity (RH)-dependencies of (**a**) d-spacing; (**b**) water adsorption capacity (at T = 23 °C); (**c**) water-vapor permeance of composite membranes; (**d**) GIWAX data at RH of ~0% and ~100% for composite membranes; (**e**) permanent gases permeance; (**f**) transmembrane pressure-dependence of water-vapor permeance (90% RH) of the composite membranes; schemes of the changes in the diffusion pathways in pure GO (**g**) and GO-intercalated with nanoribbons (**h**) and fullerenols (**i**).

**Table 1 membranes-12-00934-t001:** Microstructural parameters and statistical analysis of Raman maps for the graphene-oxide-based selective layers of the composite membranes.

Sample	D + G-Mode		Layer Thickness, nm		Estimated Porosity, %
	I, cps∙10^5^	RSD, %	SEM	TEM	
MFGO	4.33	9.5	45 ± 5	-	10
CNTGO@MFGO	1.71	8.7	37 ± 5	31 ± 3	50 ± 10
C_60_(OH)_26–32_@MFGO	3.17	22.4	30 ± 4	20 ± 3	5 ± 5

**Table 2 membranes-12-00934-t002:** Water vapor transport characteristics of composite membranes.

Sample	Permeance of Water Vapor, L∙m^−2^·bar^−1^·h^−1^		α(H_2_O/N_2_) at RH 90%	Reversible Permeance Loss at 1 Bar, %	Irreversible Permeance Loss after Pressure Increase-Decrease Cycle, %
	In a Dynamic Mode (Wet N_2_, RH~100%)	At 90% RH of Feed Stream			
MFGO	2400	63,000	47,400	58	36 *
CNTGO@MFGO	1900	51,500	5300	51	10
C_60_(OH)_26–32_@MFGO	1720	58,200	30,000	24	4.5
C_60_(OH)_26–32_@MFGO-33%	-	46,500	6642	10	2

* denotes the as-prepared sample.

**Table 3 membranes-12-00934-t003:** Diffusion and sorption coefficients of the composite membranes (at T = 23 °C, RH = 60%).

	MFGO	CNT@MFGO	C_60_(OH)_26–32_@MFGO
Thickness, nm	45	37	30
D, m^2^∙s^−1^	1.87 × 10^−9^	4.2 × 10^−10^	8.84 × 10^−10^
S, mol∙m^−3^∙Pa^−1^	4.20	6.74	2.62

## Data Availability

The data supporting the findings of this study are presented in the Appendix A and also are available upon reasonable request.

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
