# Peer review of "Oxidized Carbon-Based Spacers for Pressure-Resistant Graphene Oxide Membranes"

_membranes, 2022, doi:10.3390/membranes12100934_

Round 1

Reviewer 1 Report

This manuscript reported the effects of carbon-based spacers on the GO membrane structure and performance. By analyzing the interlayer spacing and performance under different humidity and pressure conditions, promising results were obtained. I have following concerns:

1.     What were the geometry dimensions of the nanoribbons and C60? And GO?

2.     While generally larger voids in GO layers would suffer more from compaction, why was the result in this paper showed enhanced pressure resistance by incorporation of C60 particles?

3.     How are the effects of spacer amount on GO membrane structure and performance?

Reviewer 2 Report

The article is very interesting and well written, considering a perspective and important topic. The research design is very good, the introduction clearly presents the current state of the problem and formulates the research task very clearly as well. The results are presented properly, with figures and tables helping to understand those very well. The research results support conclusions made nicely. Therefore, the article definitely deserves to be accepted to the journal as it is.

Author Response

REVIEWER 2

The article is very interesting and well written, considering a perspective and important topic. The research design is very good, the introduction clearly presents the current state of the problem and formulates the research task very clearly as well. The results are presented properly, with figures and tables helping to understand those very well. The research results support conclusions made nicely. Therefore, the article definitely deserves to be accepted to the journal as it is.

AUTHORS RESPONSE:

Dear Reviewer!

We are amazed and grateful to hear your opinion that our article deserves to be accepted to the journal as it is. We appreciate it because such recommendations become too rare in the modern reviews. Your comment inspires us to continue and deepen our research in the field of pressure stabilization of graphene oxide membranes. Thank you!

Reviewer 3 Report

The manuscript presents the preparation of graphene oxide-based membranes intercalated with fullerenols to provide stability within the interlayer space of the 2D nanosheets. The membranes were used for gas permeation tests and revealed the importance of stabilizing the GO sheets with the the use of fullerenols. The manuscript may be accepted after taking these few considerations:

1. Figures 1e-h is not clear enough to conclude the statements in lines 217-220. Is it possible to provide it using free standing GO layers? Otherwise, please explain how does this technique can show differentiation between the GO sheets and the intercalated material?

2. A shift in binding energy in the XPS spectra may indicated changes in intermolecular interactions between the materials. Please provide a comment about this.

3. Will the interlayer space between nanosheets be greatly affected by the incorporation of fullerenols? How will this affect the gas permeation property of the membrane?
